# An Enhanced FGI-GSRx Software-Defined Receiver for the Execution of Long Datasets

**DOI:** 10.3390/s24124015

**Published:** 2024-06-20

**Authors:** Muwahida Liaquat, Mohammad Zahidul H. Bhuiyan, Saiful Islam, Into Pääkkönen, Sanna Kaasalainen

**Affiliations:** Department of Navigation and Positioning, Finnish Geospatial Research Institute, 02150 Espoo, Finland; muwahida.liaquat@nls.fi (M.L.); zahidul.bhuiyan@nls.fi (M.Z.H.B.); saiful.islam@nls.fi (S.I.); into.paakkonen@nls.fi (I.P.)

**Keywords:** software-defined GNSS receiver, open-source software, parallel processing

## Abstract

The Global Navigation Satellite System (GNSS) software-defined receivers offer greater flexibility, cost-effectiveness, customization, and integration capabilities compared to traditional hardware-based receivers, making them essential for a wide range of applications. The continuous evolution of GNSS research and the availability of new features require these software-defined receivers to upgrade continuously to facilitate the latest requirements. The Finnish Geospatial Research Institute (FGI) has been supporting the GNSS research community with its open-source implementations, such as a MATLAB-based GNSS software-defined receiver `FGI-GSRx’ and a Python-based implementation `FGI-OSNMA’ for utilizing Galileo’s Open Service Navigation Message Authentication (OSNMA). In this context, longer datasets are crucial for GNSS software-defined receivers to support adaptation, optimization, and facilitate testing to investigate and develop future-proof receiver capabilities. In this paper, we present an updated version of FGI-GSRx, namely, FGI-GSRx-v2.0.0, which is also available as an open-source resource for the research community. FGI-GSRx-v2.0.0 offers improved performance as compared to its previous version, especially for the execution of long datasets. This is carried out by optimizing the receiver’s functionality and offering a newly added parallel processing feature to ensure faster capabilities to process the raw GNSS data. This paper also presents an analysis of some key design aspects of previous and current versions of FGI-GSRx for a better insight into the receiver’s functionalities. The results show that FGI-GSRx-v2.0.0 offers about a 40% run time execution improvement over FGI-GSRx-v1.0.0 in the case of the sequential processing mode and about a 59% improvement in the case of the parallel processing mode, with 17 GNSS satellites from GPS and Galileo. In addition, an attempt is made to execute v2.0.0 with MATLAB’s own parallel computing toolbox. A detailed performance comparison reveals an improvement of about 43% in execution time over the v2.0.0 parallel processing mode for the same GNSS scenario.

## 1. Introduction

The availability of Global Navigation Satellite System (GNSS) software-defined receiver (SDR) platforms have fostered research and innovation in the field of satellite navigation by enabling researchers to experiment with new algorithms, signal processing techniques, and applications. GNSS SDRs also offer greater flexibility and customization as compared to traditional GNSS receivers [1,2] by allowing developers to tailor GNSS solutions to specific requirements and integrate them into a wide range of devices and systems. GNSS SDRs also serve as a valuable tool for investigation, reproduction, validation, and cross-verification of GNSS signals and by providing open platforms to explore and share raw recorded GNSS in-phase/quadrature (I/Q) datasets [3,4,5]. These features enable GNSS SDRs to offer flexibility, adaptability, customization, and interoperability for innovation and research over hardware receivers. Advances in computing technology have also enabled a few GNSS SDRs to perform real-time signal processing [2], which is essential for applications such as transportation [6,7], aviation [8], surveying [9], and smart cities [10]. They can also adapt to different signal conditions and interference environments, leading to more robust and reliable GNSS performance. The rapid development capabilities offered by SDRs also support the proliferation of smarter hardware GNSS receivers in growing mass-consumer markets, such as smartphones, wearables, and automotive navigation systems [11,12,13], where SDRs cannot themselves compete due to cost, power, or weight restrictions.

The concept of GNSS emerged in the 1970s with the development of the U.S. Global Positioning System (GPS) and the Russian Global Navigation Satellite System (GLONASS). These systems initially used dedicated hardware receivers to process satellite signals for navigation and positioning. The 1990s saw the development of SDRs through the seminal work on real-time GNSS SDR that was first introduced by Akos in [14,15,16], which inspired a lot of research on algorithm development. A GNSS SDR can be described as a software running on a general-purpose computer translating received GNSS signal samples into a position, velocity, and time (PVT) estimate [1]. Since then, the concept has been reused by many research teams who introduced their designs focusing on signal processing algorithms, receiver architectures, and performance evaluation.

The data exchange between the various SDRs requires a certain level of standardization. This led up to the development of the Institute of Navigation (ION) SDR standard [17]. Broadly speaking, three main categories of GNSS SDRs have emerged, with all of them defined by the use of general-purpose processors to process radio frequency (RF) data from an analog front-end in essentially raw form, allowing some configuration flexibility and supporting the ION SDR standard [1]:Real-time receivers based on low-level programming languages (C or C++) (such as GRID [18,19], GNSS SDR [2], Namur [20], TUTGNSS [21], and MuSNAT [22]).Post-processing receivers written in a high-level programming language (Python/ MATLAB) for teaching and research purposes (such as FGI-GSRx [23], Borre-SDR [24,25], PyChips [17,26], softGNSS [27], and MATRIX [28,29,30]).Snapshot receivers (such as the UAB Snapshot GNSS Receiver [31,32]) optimized for very short batches of signal samples.

The availability of open-source GNSS platforms such as RTKLIB [33,34] and SDRs such as GNSS SDRs [2] and FGI-GSRx [23] has democratized access to GNSS receiver development and experimentation. These platforms have fostered collaboration, knowledge-sharing, and innovation within the GNSS community, leading to the development of new algorithms and applications. Current trends in GNSS SDRs include the integration of multiple GNSS constellations (e.g., GPS, GLONASS, Galileo, BeiDou), multi-frequency GNSS processing, enhanced security and authentication mechanisms, and the integration of GNSS with other sensor modalities for seamless navigation and localization. The ongoing development of GNSS SDRs often requires the processing of long and diverse datasets (such as datasets collected in urban areas or under adverse weather conditions), multipath mitigation, and/or interference rejection. This aids in testing the receiver key parameter indicators (KPIs) and provides enough variations for robust testing.

GNSS SDRs can also be used as an effective tool for detecting and mitigating malicious attacks aimed at disrupting or manipulating GNSS-based positioning and navigation systems, commonly known as spoofing and jamming. This may include analyzing the received GNSS signals in detail, including their characteristics, such as power levels, integrity metrics, and detecting anomalies indicative of spoofing attacks, such as the presence of counterfeit signals or discrepancies in signal properties. Techniques such as signal cross-correlation, and cryptographic authentication such as Galileo’s Open Service Navigation Message Authentication (OSNMA) [35,36,37] can be employed to ensure that the received signal is genuine. OSNMA testing on an SDR also requires the processing of longer datasets along with the need for faster processing. The Finnish Geospatial Research Institute (FGI) developed an open-source Python engine, FGI-OSNMA, for OSNMA-based navigation message authentication [38] of the Galileo E1B signal [39,40,41]. In a recent development, FGI-OSNMA has been incorporated with the FGI-GSRx software receiver to enable an OSNMA-based position authentication capability [42]. This integration will facilitate further research on the actual use of OSNMA in the context of obtaining an authentic position solution, especially in the absence of any signal- or message-level authentication mechanism with other existing GNSS systems.

To summarize, the constant evolution of GNSS, such as the inclusion of modernized signals and new services offered by the GNSS constellations, requires regular updates for GNSS SDRs. This will ensure continued functionality, security, and effectiveness in meeting users’ needs and expectations. Therefore, in this paper, we are presenting an improved and updated version of FGI-GSRx, henceforth named FGI-GSRx-v2.0.0. The major improvement in v2.0.0 is carried out by incorporating key design strategies, while focusing on one vital aspect, i.e., faster execution. These features are specifically aimed to facilitate thee processing of long datasets. This paper also presents the assessment of old and new versions of FGI-GSRx by considering some of the design aspects and KPIs specific to the assessment of the performance of software-defined GNSS receivers [43].

The rest of the paper is structured as follows: Section 2 presents an insight into the receiver architecture of FGI-GSRx-v1.0.0 and its limitations. Section 3 presents the FGI-GSRx-v2.0.0 design and architecture and a proposed parallel tracking mode. The assessment and data processing methodology is explained in Section 4. This is followed by a performance analysis of both versions of FGI-GSRx on a dataset in Section 5. Section 6 offers some concluding remarks on the current work while presenting some highlights on future work.

## 2. FGI-GSRx Software-Defined Receiver

FGI’s multi-GNSS software receiver was released as open-source in February 2022 [23] along with the book *GNSS Software Receivers* [44]. It is a MATLAB-based GNSS receiver that operates only as a post-processing receiver for raw I/Q GNSS data samples. The development of the FGI-GSRx was initiated from the work of Borre et al. [24] in 2012, in which a GNSS software-defined receiver was developed for tracking two IOV satellites (GIOVE A and GIOVE B) from the European Galileo. This was followed by the inclusion of Galileo in 2013 [45], the Chinese satellite navigation system BeiDou in early 2014 [46], the Indian regional satellite navigation system NavIC in late 2014 [47], and the Russian satellite navigation system GLONASS in 2015 [44]. The evolution of FGI-GSRx from a GPS-only receiver to a more extensive receiver supporting GNSS signals from multiple constellations (GPS, Galileo, BeiDou, GLONASS, and NavIC) offers diversity and makes it an excellent resource for researchers. The software receiver is already being used in the ‘GNSS Technologies’ course offered in several Finnish Universities.

### 2.1. Existing Sequential Receiver Architecture

The processing chain of FGI-GSRx-v1.0.0 consists of GNSS signal acquisition, code and carrier tracking, decoding of the navigation message, pseudorange estimation, and PVT estimation, as shown in Figure 1. The receiver’s salient features are listed in Table 1. As can be seen in Figure 1, the receiver allocates channels to all the acquired satellites, and then it continues to track each satellite sequentially one at a time for each time epoch until it finishes executing the last data sample. There is no dependency among the channels in this traditional tracking approach, and therefore, it would have been optimum to utilize some form of parallelism to run all the tracking channels in parallel. This parallel tracking execution strategy is considered as one of the key KPIs for faster execution of the receiver when implementing the next version of FGI-GSRx, i.e., v2.0.0. This will be further illustrated in Section 3.

### 2.2. Limitations of FGI-GSRx-v1.0.0

The current open-source version of FGI-GSRx-v1.0.0 is being used in GNSS receiver development research and in benchmarking SDR solutions in the GNSS industry. Its architecture provides a great opportunity to build and test new algorithms without the need to make extensive changes to the original code. The receiver was designed to facilitate a great deal of flexibility in terms of different key parameters at different stages of the receiver signal processing chain from acquisition to navigation. For that reason, the receiver’s memory allocation and the computing resource management was not optimal. As an example, many of the signal tracking variables were saved for the whole data length, thus consuming memory and computing resources when the data sizes grow significantly. This is especially considered a bottleneck when the processing of long datasets is needed for analysis in some certain test cases, for example, to analyze the performance of the OSNMA service of Galileo.

## 3. FGI-GSRx-v2.0.0: Design and Architecture

The limitations identified in Section 2.2 served as the motivation to suggest improvements and introduce new features in the latest release of FGI-GSRx. The latest release of FGI-GSRx, henceforth referred to as FGI-GSRx-v2.0.0, has been developed to overcome some of the limitations of the previous version by focusing on using the following design strategies. It is important to mention here that the development of v2.0.0 was carried out by optimizing and enhancing the receiver’s performance while maintaining the original design and architecture of v1.0.0. In the following, the design strategies for a GNSS SDR are introduced first before the presentation of the v2.0.0 architecture.

### 3.1. Design Strategies for Software-Defined GNSS Receivers

The evaluation process of a GNSS SDR can be broadly classified into incorporating design strategies and formulating effective assessment methods. The following subsection provides an insight into the focus areas pertaining to design strategies that were considered in the development of FGI-GSRx-v2.0.0. Several strategies can be employed to optimize the processing speed without sacrificing accuracy, functionality, usability, etc. A brief review of some of these strategies that are considered in this work is mentioned below.

***Code Profiling*****:** This refers to identifying performance bottlenecks through code profiling and optimizing critical sections of the code base. This may involve restructuring algorithms, minimizing function call overhead, and optimizing memory access patterns to improve overall processing speed. A significant effort was made in this regard for the development of v2.0.0 that included identifying the functions and processes contributing maximum processing time and removing redundant variables and computations. This effort lead to improved tracking time via the v2.0.0 sequential mode as compared to v1.0.0. A more detailed insight is presented in Section 5.1 and Section 5.2.***Memory Management and Parallelization*****:** Parallel processing techniques and efficient memory management of the processing unit can minimize memory access latency and improve overall processing speed. The FGI-GSRx-v2.0.0 parallel processing mode is aimed at distributing computational tasks by invoking multiple MATLAB instances so as to utilize the maximum processing power of the processing units, such as CPU cores. This can significantly speed up signal processing algorithms, especially those that are inherently parallelizable, such as correlation and FFT (Fast Fourier Transform) operations. An analysis on the performance of memory management using various versions of FGI-GSRx is shown in Section 5.3.

### 3.2. FGI-GSRx-v2.0.0 Receiver Architecture

FGI-GSRx-v2.0.0 was developed to overcome existing shortcomings in v1.0.0 by focusing on the design strategies presented in Section 3.1. Figure 2 presents the data flow of the v2.0.0 architecture, which is similar to v1.0.0. The three main blocks (*Acquisition, Tracking, and Navigation*) offered by v2.0.0 inherently follow the same functional methodology as v1.0.0, with the exception of the tracking block. A brief description of these blocks is presented below.
***Acquisition***: In the first step, the acquisition block searches and acquires satellite signals one at a time, which is followed by fine acquisition in the second stage. A detailed explanation of second stage acquisition is presented in Section 3.3. The signals are then handed over to the tracking block once the search is complete. v2.0.0 also supports multiple GNSS constellations like v1.0.0. The acquisition data contains the information regarding each signal acquired (observations, duration, signal, channel) as well as statistics from the acquisition phase (e.g., peak metric, peak value, variance, baseline, standard deviation, code phase, carrier frequency, and satellite ID).***Tracking***: The tracking block utilizes the acquisition data and correlates the incoming signal with signal replicas to generate tracking measurements from all signals and satellites. In v1.0.0, the signal tracking of each satellite is processed sequentially, which was not very effective, especially for the processing of longer datasets leading to higher processing time. To overcome this limitation, in v2.0.0, the user can conveniently choose between two possible options to execute the tracking process. These include:–***Sequential Tracking Mode***: This mode is developed on the similar lines as v1.0.0. However, v2.0.0 offers an improved sequential tracking architecture, which is carried out by code profiling, efficient handling of data variables, and identifying and optimizing high complexity and time-consuming processes that ultimately contributed to faster data processing.–***Parallel Tracking Mode***: The parallel tracking mode is aimed at maximizing the CPU utilization to speed up the processing. The parallel processing is conducted at the signal tracking stage only since this is the most computationally extensive part of the receiver. The idea is to initiate multiple parallel instances of MATLAB for processing each satellite individually to achieve maximum CPU utilization from the processing platform. In this particular case, the operating system takes care of running multiple instances by allocating enough resources for these demanding parallel instances. For example, if there are, altogether, 17 GPS and Galileo satellites available from the acquisition stage, the new receiver architecture will open 17 MATLAB instances where each instance will process one individual tracking channel. The work flow diagram of the v2.0.0 parallel tracking block is divided into three main steps, as shown in Figure 3.***Multi-GNSS Navigation***: This block offers data decoding of the output generated from the tracking block and converts the processed track data into receiver observables in terms of satellite-specific pseudoranges and ephemeris for each GNSS constellation. The navigation block has four main tasks: to convert the measurements into observations, calculate each satellite’s PVT, apply satellite and environmental corrections, and, finally, estimate the PVT solution of the user with the corrected pseudoranges.

**Figure 3 sensors-24-04015-f003:**
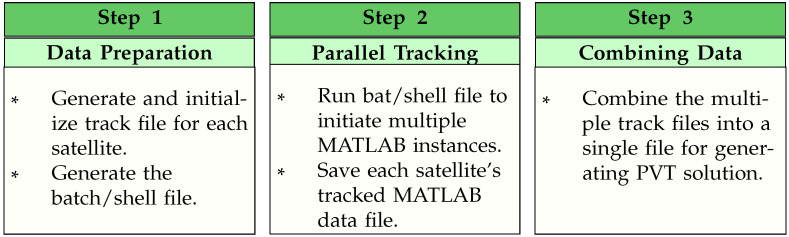
FGI-GSRx-v2.0.0 parallel tracking mode work flow.

#### FGI-GSRx-v2.0.0 Parallel Tracking Mode

The detail settings and procedure to execute each step is described in the user manual (for both Windows and LINUX platforms), which is available for download from the FGI-GSRx-v2.0.0 online repository [48]. It is pertinent to mention here that while the parallel tracking mode speeds up the processing, the user has to manually go through all steps of the parallel processing listed above, which adds complexity to the usability of the receiver.

### 3.3. Other Functional Enhancements

All the modifications in FGI-GSRx-v2.0.0 from v1.0.0 are highlighted in the v2.0.0 release note [48]. Most of these modifications are on the source codes that deal with the optimization of memory and computing resources. Apart from that, the most significant functional update is in the signal acquisition, where a two-stage acquisition is introduced. The two-stage acquisition algorithm is very effective in mitigating the correlation peak splitting effect caused by the presence of bit sign transitions in the signal segments, in particular, with modern GNSS signals where bit transition may occur within the data blocks during the traditional Fast Fourier Transform (FFT)-based acquisition [49].

In the case of a successful first step acquisition of a particular satellite, the receiver proceeds with a second acquisition stage for the fine estimation of the carrier doppler of the acquired satellite. The receiver obtains the code phase and the carrier doppler of the acquired satellite from the first stage as input to the second stage. At the second stage, the receiver advances the code phase of the satellite, so it starts the correlation of the received signal with the local replica code from the beginning of the code epoch, thus avoiding any possibility for bit transition within the data block. In addition to that, the receiver also down-converts the carrier doppler from few hundreds of Hertz (Hz) to within ±25 Hz by appropriately tuning the frequency step in the FFT-based search space. A representative two-stage acquisition plot is shown in Figure 4. It can be seen that the two-stage acquisition correctly estimates the carrier Doppler for Galileo PRN 34 at 1725 Hz, whereas after the first stage acquisition, the estimated Doppler is 1625 Hz. This acquisition result was obtained with a FGI-GSRx software receiver with 4 milliseconds (ms) of coherent integration and 2 blocks of non-coherent summation in order to show the advantage of two-stage acquisition to solve the problem of bit sign transitions within the acquisition signal segments. With this above acquisition configuration (i.e., 4 ms of coherent integration and 2 blocks of non-coherent summation), FGI-GSRx-v1.0.0 acquires Galileo PRN 34 with 1625 Hz, thus leading to bit error at the navigation stage due to not locking to the right carrier frequency. This phenomenon can be avoided with two-stage acquisition in FGI-GSRx-v2.0.0.

## 4. Assessment and Data Processing Strategy

This section provides the data processing and assessment strategy utilized to provide a common framework for an unbiased and effective performance comparison of FGI-GSRx-v1.0.0 and FGI-GSRx-v2.0.0.

### 4.1. Assessment Strategies

To assess the performance and development of v2.0.0, some key performance indices (KPIs) are considered. These assessment methods are aimed at providing a platform for measuring the improvement and can be considered as a valuable tool for bench-marking [50]. To streamline this process, a generative list of 16 KPIs related to GNSS SDRs is presented in [43]. It is also impractical to expect an analysis against all potential measures and published methods. Therefore, for the development of v2.0.0, a subset of six points were selected, as they are relevant to our scope and are explained in Table 2.

### 4.2. Data Processing Strategy

To ensure a fair comparison, all the simulations were run on a dedicated processing unit, whose details are given in Table 3.

The reference dataset used in this paper was recorded as an open sky signal captured by Septentrio’s PolarNt Choke Ring antenna at FGI, Finland. The true receiver position is at 60.182° N, 24.828° E with an altitude of 47.248 m (m). A dual-band RF front-end from Nottingham Scientific Limited (NLS) was used to down-convert the captured GNSS signal into the raw I/Q format required for processing by FGI-GSRx. The details of the reference dataset are given in Table 4. The signal settings (or user requirements) used in this analysis are the same for all versions of FGI-GSRx and are presented in Table 5, where non-coherent normalized early minus late (NNEML) was used as the DLL discriminator function. A more detailed explanation of the receiver’s acquisition and tracking techniques can be found in [44]. Figure 5 illustrates the sky plot for both GPS and Galileo satellites recorded on 30 October 2023, at around 14:19:06 UTC. There are 11 GPS and 6 Galileo satellites acquired with both versions of FGI-GSRx. The raw I/Q dataset is publicly available in [51].

## 5. Result Analysis

The development of FGI-GSRx-v2.0.0 was carried out by considering the design strategies (Section 3.1) while focusing on improving the design KPIs (Section 4.1) to overcome the limitations of v1.0.0. All the versions of FGI-GSRx are programmed in the MATLAB environment. The underlying design methodology is the same for both versions. However, some functions are optimized for faster and accurate processing in the later release. The source code for both versions of FGI-GSRx is supported by the book *GNSS Software receiver*, along with raw GNSS data files, default user settings, and processed data files. That means that the user could also utilize v2.0.0 for processing the open data files utilized in *GNSS Software receiver*. v2.0.0 is also supported by an additional user manual to incorporate the parallel processing mode (addressing both Windows and LINUX settings). This section presents a detailed insight and discussion into the performance comparison of both versions of FGI-GSRx through various aspects.

### 5.1. Code Profiling

To speed up the processing, code profiling was conducted to identify the functions and processes that were contributing to the maximum processing time. These functions were reviewed and optimized by reducing unnecessary computations, variables, and memory access. An insight into some of the functions contributing to the maximum run time, particularly from the tracking block, is presented in Table 6. These functions were modified in the new version, and the resultant improvement achieved by v2.0.0 with respect to v1.0.0, in terms of the time taken to run each function, is also discussed in Table 6. v2.0.0 also offers some new functionalities, with the addition of the parallel tracking mode and multicorrelator processing. The detailed description of modifications and new additions are not in the scope of this paper; however, they are available in the FGI-GSRx-v2.0.0 release note [48] and user manual for a more thorough insight for interested readers.

### 5.2. Data Processing

The FGI-GSRx receiver architecture allows us to analyze the *efficiency* of the data processing blocks as a modular structure. It gives users the flexibility to evaluate three core blocks (*Acquisition, Tracking, and Navigation*) individually as well as collectively. Taking advantage of this functionality, Table 7 presents an insight into the run time accumulated by each block in different versions of FGI-GSRx. The acquisition block of v2.0.0 offers an extra step to offer fine acquisition, which is the reason for the higher acquisition time by v2.0.0, which was explained in Section 3.3. For the presented dataset, the navigation processing time for v2.0.0 is slightly higher than v1.0.0 mostly due to the processing of a higher number of satellites in the PVT computation, although the navigation block offers the same code functionalities for both versions. More insight into this is presented in Section 5.4. The maximum run time is consumed by the tracking block in all the competing cases. This observation also served as the inspiration to introduce the parallel processing mode for v2.0.0. It can be observed that v2.0.0 reduces the total processing run time by approximately 40% as compared to the previous version. The added feature of a parallel processing block further improves the total simulation run time by approximately 59% for the given dataset.

### 5.3. Resource Management

FGI-GSRx-v2.0.0 specifically aims at improving the processing time, which is vital for the processing of long datasets. The processing unit (or CPU) utilization for both versions of FGI-GSRx is presented in this section to analyze the resource management and utilization. It is important to mention here that to ensure a fair comparison, the processing unit was only processing FGI-GSRx during the recording of the CPU usage.

Figure 6 presents the CPU utilization for both versions using the sequential tracking mode using all satellites. On average, approximately 21% of CPU resources were utilized during the entire simulation interval for both versions. v2.0.0 offered a more uniform distribution of resources as compared to v1.0.0. Therefore, it can be concluded that in terms of *Resource Management*, the optimization of core functions enabled the much-improved usage of the processor by v2.0.0. It was previously highlighted in Table 7 that the maximum processing time is utilized by the tracking block of the receiver. This motivated us to develop the parallel processing block for processing only the tracking block via v2.0.0 by utilizing the CPU resources more effectively. Figure 9a shows the CPU usage while processing the v2.0.0 parallel processing mode. The average CPU usage is approximately 85%,which is significantly higher than the sequential processing mode. This is also the reason for the significant improvement in overall processing time by v2.0.0, as presented in Table 7.

#### Comparison with MATLAB Parallel Computing Toolbox

MATLAB also offers a similar parallel processing feature via a parallel computing toolbox (PCT). A parallel computing toolbox takes advantage of the available computer resources by distributing tasks and executing them in parallel. It accelerates the code by creating a parallel pool of MATLAB workers using interactive parallel computing tools, such as *parfor* and *parfeval*, to provide automatic parallel support. For the default local profile, the default number of workers is one per physical CPU core using a single computational thread. The processing unit used in this analysis is supported by 14 CPU’cores.

Table 8 presents a comparison of performance for the v2.0.0 parallel mode and with the v2.0.0 sequential mode run with MATLAB PCT for a variable number of satellites. For the processing of six Galileo satellites, the CPU usage by both receivers is shown in Figure 7. The processing time is almost the same for both entities; however, v2.0.0 parallel has a higher average value of 69% as compared to 38% by MATLAB PCT. However, when we processed the GPS-only constellation of 11 satellites, MATLAB PCT showed an improved performance of 30% while keeping the average CPU usage at a lower rate of 58%, as compared to v2.0.0 parallel mode’s average CPU usage of 90%, as presented in Figure 8.

A similar trend is also observed when we processed both GPS and Galileo satellites (a total of 17 satellites). As opposed to v2.0.0 where 17 MATLAB instances were initiated simultaneously for processing, PCT initially facilitates 14 channels and accommodates the remaining channels accordingly. This can be seen in Figure 9b, where the processing is conducted in a more distributed manner as compared to thhe v2.0.0 parallel mode performance presented in Figure 9a. This limitation is hardware-driven and may result in faster or slower processing based on the number of available cores of the processing unit. Figure 9 presents the recorded CPU utilization while processing v2.0.0 with the MATLAB PCT for this scenario. FGI-GSRx with MATLAB PCT offers an average CPU usage of 60%, which is lower than the v2.0.0 parallel processing average, but it resulted in a faster processing time (43% improvement over the v2.0.0 parallel tracking mode).

To summarize, MATLAB PCT offers improved performance as compared to thhe v2.0.0 parallel mode, especially when the number of satellites to be processed is higher. The superior performance of MATLAB PCT could be accounted for by the optimized resource management offered by the MATLAB inbuilt functions. It is also important to mention here that the PCT toolbox is a paid toolbox offered by MATLAB, whereas the v2.0.0 parallel mode does not depend on the PCT toolbox.

### 5.4. Accuracy

To investigate the *accuracy* assessment criteria, an insight into the stand-alone static position accuracy is presented. For the presented datasets, 11 GPS and 6 Galileo satellites were acquired in the acquisition stage by both versions of FGI-GSRx. Figure 10 shows the obtained position solution by both versions of FGI-GSRx using the sequential processing mode. The positioning solutions obtained by both versions used the ionosphere final product from IGS (International GNSS Service), which is provided in IONEX (Ionosphere Exchange) format [52]. This is also followed by a detailed analysis presented in Table 9 that includes indices for position solution, such as 3D root mean square (3DRMS) error, dilution of precision (DOP), horizontal and vertical position deviations, etc. It is pertinent to mention here that both versions of FGI-GSRx tracked the same number of satellites and offer similar functionalities at the navigation level. An improved preamble detection algorithm in the frame decoding block enabled v2.0.0 to process the ephemeris of GPS PRN 27 and Galileo PRN 21 successfully. Therefore, the availability of a higher number of satellites in this particular case resulted in better position accuracy for v2.0.0 than v1.0.0.

## 6. Conclusions

This paper presents the latest update of the FGI’s open-source GNSS software-defined receiver ‘FGI-GSRx’ while analyzing the performance of its new receiver tracking architecture. In particular, the FGI-GSRx-v2.0.0 release offers improvements in two main areas: i) functional optimization, and ii) faster processing at the signal tracking stage compared to the previous version by introducing parallel tracking architecture as an added tracking option. The improved performance of v2.0.0 was verified by assessing a few pertinent key performance indices and bench-marking its performance using the MATLAB parallel computing toolbox. Additionally, v2.0.0 offers two-stage acquisition and an optimized performance as compared to v1.0.0. The usability of the new architecture is slightly more complex for parallel processing than sequential processing. However, it facilitates faster processing of the receiver’s tracking block, which is a significant advantage, especially for the processing of longer datasets, which is a necessary requirement to facilitate cryptographic authentications, such as Galileo’s Open Service Navigation Message Authentication. It is also important to mention that the source code for v2.0.0 and the dataset presented in this paper are shared as open-source by the National Land Survey of Finland.

## Figures and Tables

**Figure 1 sensors-24-04015-f001:**
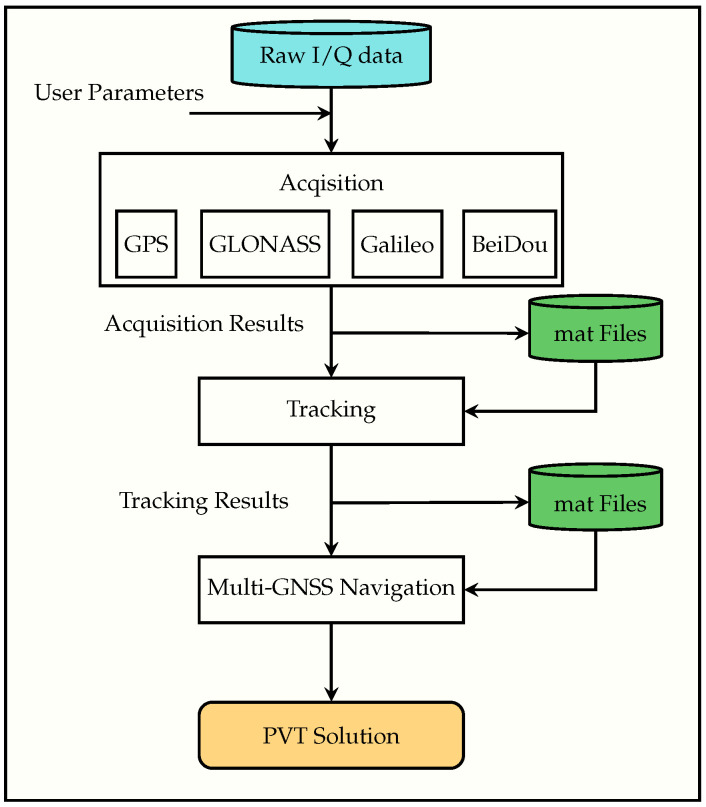
FGI-GSRx sequential architecture. The green parts indicate the option to use a pre-stored output from acquisition and tracking.

**Figure 2 sensors-24-04015-f002:**
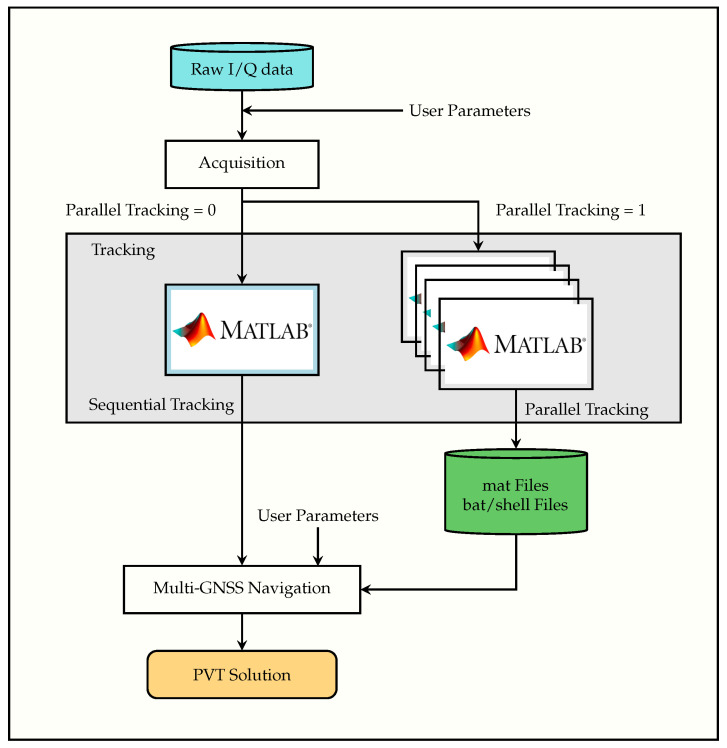
FGI-GSRx-v2.0.0 architecture. The green parts indicate the option to use a pre-stored output from acquisition and tracking.

**Figure 4 sensors-24-04015-f004:**
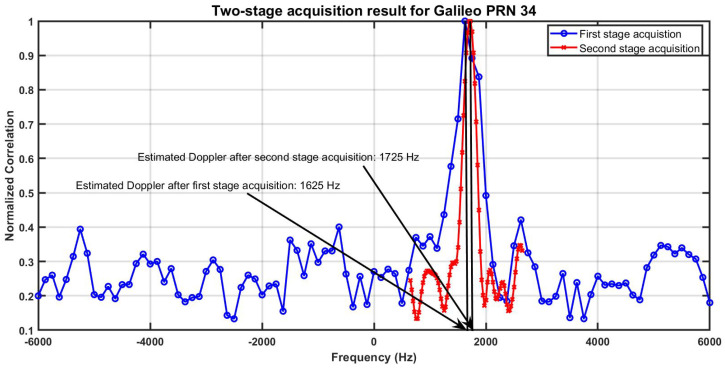
FGI-GSRx-v2.0.0 two-stage acquisition.

**Figure 5 sensors-24-04015-f005:**
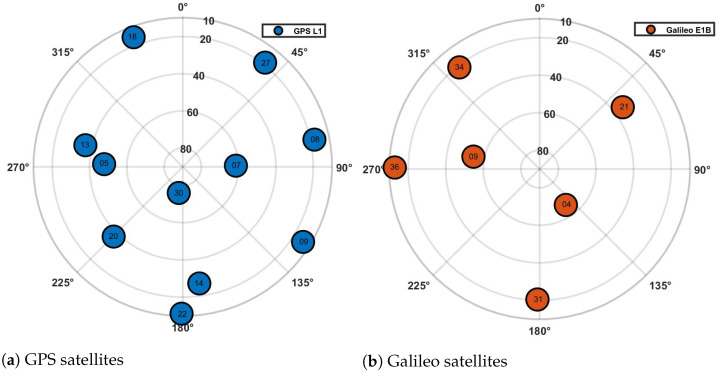
Sky plots for GPS and Galileo satellites at the beginning of data collection.

**Figure 6 sensors-24-04015-f006:**
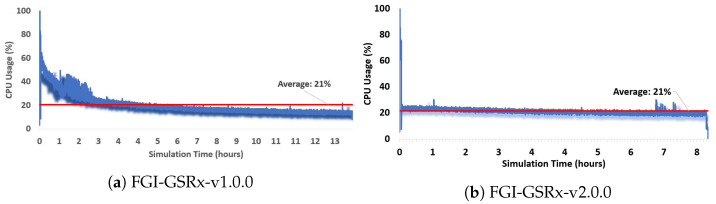
Processor usage utilization for the entire simulation interval for the sequential processing mode of FGI-GSRx.

**Figure 7 sensors-24-04015-f007:**
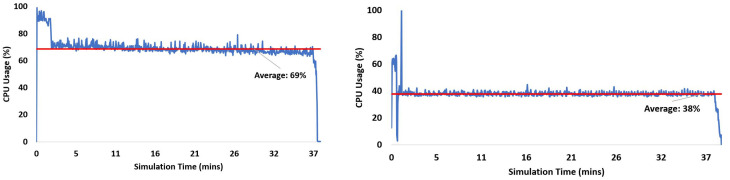
Galileo only: CPU usage for signal tracking of the v2.0.0 (**left**) Parallel processing mode. (**right**) Sequential processing with MATLAB parallel computing block.

**Figure 8 sensors-24-04015-f008:**
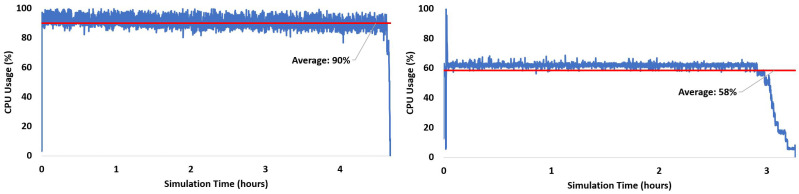
GPS only: CPU usage for signal tracking of the v2.0.0 (**left**) Parallel processing mode. (**right**) Sequential processing with MATLAB parallel computing block.

**Figure 9 sensors-24-04015-f009:**
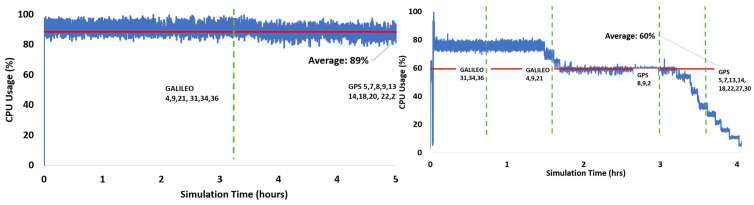
GPS and Galileo: CPU usage for signal tracking of the v2.0.0 (**left**) Parallel processing mode. (**right**) Sequential processing with MATLAB parallel computing block.

**Figure 10 sensors-24-04015-f010:**
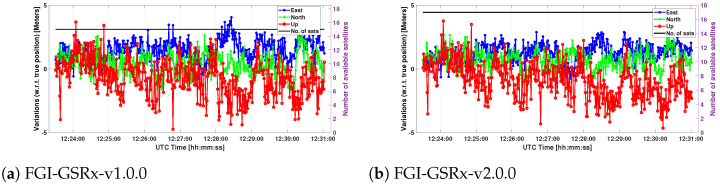
Position deviation plots generated by FGI-GSRx.

**Table 1 sensors-24-04015-t001:** Main features of FGI-GSRx.

Feature	Solution	Remark
Operating system	Windows10, LINUX	Supports any operating system that can host MATLAB or OCTAVE.
Programming environment	MATLAB	MATLAB 2019 or later versions and OCTAVE.
Input source	Raw I/Q data	Digitized raw I/Q samples after analog-to-digital conversion.
Processing mode	Post-processing	Can process raw I/Q data or load previously processed and saved acquisition or tracking MATLAB data files.
Supported GNSS	GPS L1, Galileo E1B, BeiDou B1, GLONASS L1, NavIC L5	Listed signals are for open-source only. In-house version of FGI-GSRx can process additional signals.
Acquisition	FFT-based signal acquisition	The receiver searches for all the listed satellites defined by the user in the configuration file.
Tracking	Three stage tracking	(i) Pull in, (ii) Coarse tracking, (iii) Fine tracking.
Navigation	Least Square Estimation (LSE)-based position computation	Possibility of selecting satellites based on Carrier-to-Noise density ratio ($C/N_{0}$) and elevation cut-off mask.

**Table 2 sensors-24-04015-t002:** KPI compliance FGI-GSRx.

KPI	Remarks
Portability	It refers to the usability of the same software in different environments. Both versions can be executed on any platform supporting MATLAB, e.g., LINUX and Windows).
Openness	This refers to the degree to which something is accessible to view, modify, and use. The source codes of both versions of FGI-GSRx are publicly available and can be modified as per user requirement [23].
Interoperability	This refers to the possibility to exchange information with other free and proprietary software, devices, sensors, or systems. For example, the ongoing work in [42] is a good example of integration of FGI-GSRx with FGI-OSNMA.
Reproducibility	It describes the capacity of a whole process to be replicated either by own team or some external research group. Both versions of FGI-GSRx support this KPI and have been tested thoroughly by an in-house research team, while v1.0.0 has also been tested by researchers [3,45,46].
Usability	This KPI is concerned with the availability of a (versioned) user manual, tutorials, and detailed procedures. In this context, both versions are supported by datasets, release notes, and user manuals, which are available for download from the online repository [48].
Efficiency	It refers to optimizing the speed and memory requirements or power consumption by the processor running the SDR. FGI-GSRx-v2.0.0 offers better efficiency as compared to v1.0.0. More insight into this functionality is presented in Section 5.2 and Section 5.3.
Accuracy	Both versions offers similar code functionalities at the navigation level. However, the minor difference in positioning accuracy can be contributed to the fewer satellites processed by v1.0.0 than v2.0.0 for the given dataset in Section 5.1 and Section 5.4.

**Table 3 sensors-24-04015-t003:** Processing unit specifications.

Processing Unit	
Processor	12th ${GenINTEL}^{®}Core^{TM}$ i7-12700H*x*20
RAM	32 GB
Operating System	64 bit Ubuntu 20.04.6 LTS

**Table 4 sensors-24-04015-t004:** Reference data specifications.

Reference Dataset	
Date	31 October 2023
Time	≈12:23 (UTC)
Duration	460 s
Size	≈11.4 GB
Location	FGI rooftop, Espoo
Receiver dynamics	Static
Antenna	Septentrio’s PolaNt Choke Ring
GNSS front-end	NSL Stereo dual band

**Table 5 sensors-24-04015-t005:** Signal settings for data processing.

Signal Configurations		
Center frequency	1569 MHz	
Bandwidth	4.2 MHz	
Sampling frequency	26 MHz	
Quantization	8 bits	
I/Q	Complex	
GNSS constellations	GPS	Galileo
**Acquisition**		
Visible Satellites	PRN 5, 7, 8, 9, 13, 14, 18, 20, 22, 27, 30	PRN 4, 9, 21, 31, 34, 36
Replica modulation	BPSK	CBOC
Max. search freq.	7000 Hz	6000 Hz
Coherent Integration	5 ms	4 ms
Non-coherent Integration number	5	3
Threshold	9	9
**Tracking**		
**DLL**		
Discriminator	NNEML	NNEML
Correlator Spacing	0.1	0.1
Damping ratio	0.7	0.7
Noise bandwidth	1	1
**FLL**		
Discriminator	atan2	atan
Wide bandwidth (Hz)	100	75
Narrow bandwidth (Hz)	50	45
Very-narrow bandwidth (Hz)	10	5
Damping ratio	1.8	1.5
Loop gain	0.7	0.7
**PLL**		
Wide bandwidth	15	15
Narrow bandwidth	15	15
Very-narrow bandwidth	10	10
Damping ratio	0.5	0.7
Loop gain	0.2	0.2

**Table 6 sensors-24-04015-t006:** Comparison of run times for multiple functions in different versions of the serial processing mode of FGI-GSRx.

Function Name	Function Description	Run Time (v1.0.0) (hh:mm:ss)	Run Time (v2.0.0) (hh:mm:ss)	Improvement (%)
‘gsrx’	Main function to execute the whole process chain of FGI-GSRx.	14:00:10	08:24:14	40
‘doTracking’	Main tracking function to conduct code and carrier tracking.	13:56:20	08:19:00	40
‘GNSSTracking’	Performs state-based tracking for the received signal.	06:25:10	03:16:21	49
‘GNSSCorrelation’	Performs code and carrier correlation.	07:26:33	04:46:12	36
‘carrierMixing’	Performs carrier and code mixing.	04:09:03	01:53:03	55
‘CN0fromSNR’	Function for estimating CN0 values using SNR.	01:31:05	01:07:56	25
‘phaseFreqFilter’	Loop filter to conduct carrier tracking.	01:06:12	00:44:41	32
‘getDataForCorrelation’	Read data for processing.	00:51:16	00:29:25	43

**Table 7 sensors-24-04015-t007:** Simulation run time comparison analysis of FGI-GSRx-v2.0.0 with respect to FGI-GSRx-v1.0.0.

Function	FGI-GSRx-v1.0.0	FGI-GSRx-v2.0.0 Sequential		FGI-GSRx-v2.0.0 Parallel	
	Processing Time (hh:mm:ss)	Processing Time (hh:mm:ss)	Improvement (%)	Processing Time (hh:mm:ss)	Improvement (%)
Acquisition	00:02:09	00:02:14	–1.8	00:02:14	–1.9
Tracking	13:56:20	08:19:00	40.31	05:42:44	59.2
Navigation	00:01:40	00:02:00	–16.6	00:02:00	–16.6
Total Run Time	14:00:10	08:24:14	39.98	05:47:53	59.13

**Table 8 sensors-24-04015-t008:** CPU usage comparison analysis of the v2.0.0 parallel tracking mode with MATLAB PCT.

Constellation	No. of Satellites	FGI-GSRx-v2.0.0 Parallel		FGI-GSRx-v2.0.0 with MATLAB PCT		
		Processing Time (hh:mm:ss)	CPU Average (%)	Processing Time (hh:mm:ss)	CPU Average (%)	Improvement w.r.t v2.0.0 Parallel (%)
Galileo only	6	00:37:56	69	00:37:48	38	0.35
GPS only	11	04:07:25	90	02:51:24	58	30
GPS + Galileo	17	05:42:44	89	03:14:36	60	43

**Table 9 sensors-24-04015-t009:** Position solution accuracy computed by FGI-GSRx.

	FGI-GSRx-v1.0.0	FGI-GSRx-v2.0.0
**Available Satellites for Position Solution**		
GPS	10 Ephemeris for GPS PRN 27 is not found.	11
Galileo	5 Ephemeris for Galileo PRN 21 is not found.	6
**Horizontal Position (m)**		
Error50	1.74	1.62
Error95	3.06	2.65
Max	4.39	3.83
Std. Dev.	0.76	0.62
RMS	2.09	1.95
Mean	1.76	1.62
**Vertical Position (m)**		
Error50	1.04	1.11
Error95	3.19	3.19
Max	4.7	4.65
Std. Dev.	0.98	0.99
RMS	1.37	1.39
Mean	1.25	1.32
**DOP**		
Pdop	1.20	1.13
**3DRMS (m)**	2.49	2.40
**No. of Positioning Epochs**	451	451

## Data Availability

FGI-GSRx is an open-source software-defined GNSS receiver and is available for download from https://github.com/nlsfi/FGI-GSRx (accessed on 29 April 2024).This release is also supported by release notes and a detailed user manual to facilitate the users with the configuration and usability of FGI-GSRx. The GNSS data captured at the Otaniemi premises of Finnish Geospatial Research Institute (FGI) in Espoo, Finland, on 31 October 2023, and mat files generated by both versions of FGI-GSRx discussed in this paper are publicly accessible at https://doi.org/10.23729/9559efea-22fc-48ac-8de4-c4d1cba367be (accessed on 30 April 2024).

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
