# Peer review of "An Enhanced FGI-GSRx Software-Defined Receiver for the Execution of Long Datasets"

_sensors, 2024, doi:10.3390/s24124015_

Round 1
Reviewer 1 Report
Comments and Suggestions for Authors
Review Comments on the manuscript titled as An Enhanced FGI-GSRx Software-Defined Receiver for the Execution of Long Datasets.
Thank you for submitting the manuscript entitled as An Enhanced FGI-GSRx Software-Defined Receiver for the Execution of Long Datasets. Overall, this paper demonstrates significant advances in GNSS receiver technology, particularly in terms of positioning accuracy and processing speed. The following are some specific comments and suggestions:
The paper is well organized with a logical flow from receiver architecture and design improvements to performance evaluation. The new version of FGI-GSRx introduces parallel processing architecture and signal tracking optimization, significantly improving processing speed and performance, which is crucial for the development of GNSS receivers. The author provides open source code and data sets to facilitate other scholars to verify and expand research.
Major Comments:
1. It is recommended to provide a more detailed description of the implementation details of parallel processing architecture, including its performance and resource management strategies on multi-core processors.
2. The paper provides a comparison of various performance indicators, but some data analysis lacks depth. A more detailed explanation of performance under different conditions, such as urban environments or severe weather, is recommended.
3. Further explanation of the specific methods and parameter settings used during data processing will help other researchers replicate the experimental results.
Minor Comments:
1. Although the paper mentions the superior performance of MATLAB PCT, it lacks comparative analysis. In particular, it is necessary to conduct a comparative analysis with the well-known GNSS-SDR, which is widely used in the industry and has achieved many successes.
2. With the development of GNSS-R and GNSS-RO technology, weak GNSS signals are captured through open-loop tracking and used for atmospheric and ocean information acquisition. Many GNSS-SDR software does not have relevant open interfaces or implementations in open-loop technology. Of course, it may also involve transplanting the processing algorithm to C++ or GPU platform. Perhaps the author can briefly introduce the expansion and planning of FGI-GSRx in this area.
Overall, the manuscript does an excellent job of presenting the improvements in FGI-GSRx v2.0.0, demonstrating significant enhancements in performance and processing speed. However, adding detailed performance analysis, practical application discussions, and technical descriptions will further improve the quality of the paper. I look forward to seeing these suggestions addressed in the revised version.
Comments on the Quality of English LanguageThe paper is well-organized, with a logical flow from receiver architecture and design improvements to performance evaluation. The text in some figures (e.g., Figures 7 to 10) are too small, making the legend and axes look unclear. Enhancing the quality of these figures will help readers better understand the data presented.
Author Response
We would like to thank the reviewer for the comments. Please see our responses as an attachment.

Reviewer 2 Report
Comments and Suggestions for Authors
This update of the FGI’s open-source GNSS SDR FGI-GSRx is presented. The receiver can be used for a series of satellite navigation tasks. The paper can published after a minor update. A general recommendation is to provide more specific applications where SDR solutions can be useful. Do they include remote sensing?
Specific Remarks:
L. 50. A GNSS SDR can be described as a software running on a general-purpose computer translating received GNSS signal samples into a position, velocity, and time (PVT) estimate [2].
Also, important for applications in remote sensing techniques, such as radio occultation, are these: pseudorange, signal-to-noise ratio, and phase.
L. 56. ION
Explain this and the other abbreviations at the first instance.
L. 250. In addition to that, the receiver also refines the carrier doppler from few hundreds of Hertz (Hz) to within ±25 Hz
It is better to say that the receiver down-converts the Doppler.
Table 5. PLL/GPS, discriminator = atan
It is a two-quadrant detector getting rid of navigation bits, right?
Figure 10. What do E, N, and U refer to?
Comments on the Quality of English Language
L. 21. A detail[ed] performance comparison
L. 36. Advances in computing technology _has_ also enabled
have
L. 78. This aids in towards testing the performance of a set of receiver key parameter indicators (KPIs) and enables GNSS receivers to incorporate enough variations within the dataset so that it can be trusted for robust testing.
Simplify the language, e.g.: This aids in testing the receiver key parameter indicators (KPIs) and provides enough variations for robust testing.
L. 179. can lead to minimize
can minimize
L. 271. In a bid to streamline this process
To streamline this process
L. 276. In order to ensure fairness in comparison
To ensure a fair comparison
L. 296. some functions are optimized to cater for faster and accurate processing
some functions are optimized for faster and accurate processing
L. 305. In the bid to execute faster processing
To speed up processing
L. 342. Recourse Management
Resource Management
L. 347. The average CPU usage can be seen to be approx. 85%
The average CPU usage is approx. 85%
Author Response
We want to thank the Reviewer for the comments. Please see attached for our responses.

Reviewer 3 Report
Comments and Suggestions for Authors
The authors present the new version of their FGI-GSRx tool, with improved execution speed. The paper is well written and reads easily. I only have a few suggestions.
line 159: the word "integrity" is not clear to me here. What does it refer to?
In section 3.3, the second acquisition stage is explained, but nothing is said about the first stage. A short paragraph explaining the principle of the first stage would help to put the second stage in context and better understand its role.
On line 251: “by appropriately tuning the frequency”. I’d suggest adding a few words to explain what is behind “appropriately”. How is this tuning done?
Section 5.3.1, it is not clear to me in what mode the FGI-GSRx tool was configured when using the Matlab PCT toolbox. Was the FGI-GSRx configured in the sequential mode? This should be clarified.
Line 379: “is also dependent on the number of available cores”. This is presented as a drawback of Matlab PCT, but the same remark applies to the v2.0.0 parallel mode. Maybe this sentence should be removed.
Line 395: “The reason … is the availability of higher number of satellites in case of v2.0.0 than v1.0.0”. This contradicts line 386 where it is said that both versions acquired the same number of satellites. It would help to add a short note explaining why more satellites are processed in the v2.0.0 position solution while the same number of satellites are acquired (and I guess tracked?) in both versions.
Some typos:
line 115: software define receiver (d missing)
line 126: galieo
line 315: through -> thorough?
Line 342: recourse -> resource
Author Response

(The authors gave the same response as above.)

Reviewer 4 Report
Comments and Suggestions for Authors
(1) A brief summary (one short paragraph) outlining the aim of the paper
and its main contributions;
Paper deals with development of the MATLAB-based GNSS software-defined receiver ’FGI-GSRx’ (version 2.0.0). Authors present an updated version of FGI-GSRx, which is also available as an open-source resource for the research community.
Stress is done on the improved performance as compared to its previous version, especially for the execution of long datasets.
The main achievements are optimization the receiver’s functionality and adding the parallel processing feature to ensure faster capabilities to process the raw GNSS data.
As a overall result is about 40% run time execution improvement over FGI-GSRx-v1.0.0 and about 59% improvement in case of parallel processing mode with 17 GNSS satellites from GPS and Galileo.
(2) Broad comments highlighting areas of strength and weakness. These comments should be specific enough for authors to be able to respond;
It is clear, that authors did reached the main goal by developing the new version of GNSS software-defined receiver. However the comparisons of all technical achievements are done against the previous version of the same receiver FGI-GSRx-v1.0.0. By the head of reviewer, it is not enough. The comparisons against developments of other authors should be done.
(3) Specific comments referring to line numbers, tables, or figures.
Tables and figures are fine. Just in Table 9 the dimensions should be added for each presented parameter.
(4) Are the references appropriate?
List of references is very good, however reference to book [47] should be corrected:
Borre, K., Fernández-Hernández, I., López-Salcedo, J.A., Bhuiyan, M.Z.H. (2022) GNSS Software Receivers. Cambridge University Press.
(5) Overall conclusion
Scientific soundness is average, practical findings are important and promising.
In general I like presented paper, it gives a broad introduction on achievements of other authors, gives the convincing practical results.
Paper is written in good manner, well structured and could be printed in "Sensors".
Author Response

(The authors gave the same response as above.)

Round 2
Reviewer 1 Report
Comments and Suggestions for Authors
I don't think the authors have done a good job of explaining or supplementing the content I reviewed.
The article titled An Enhanced FGI-GSRx Software-Defined Receiver for the Execution of Long Datasets does not well express the upgrade of FGI-GSRx software from the perspective of principle, function expansion, and application scenario enhancement.
The relevant experimental design follows the relevant papers of FGI-GSRx v.1.0, but it is not well connected and briefly explained in this article.
The algorithm uses a new parallel method to achieve fast acquisition of GNSS signals, but as far as I know, it has been implemented in GNSS-SDR (and uses the CPP language with higher computing performance), and supports multi-system multi-frequency signal acquisition. This software is more of a teaching example software, rather than a pioneering progress in GNSS weak signal tracking or GNSS integrity assurance under abnormal conditions.
In the first review, I mentioned that there are many aspects that can be continued in the research content, but the author's response was too perfunctory, and there was no relevant update in the paper.
Author Response
Please find our responses in an attachment.
